# Impact of Microbiota Diversity on Inflammatory Bowel Disease

**DOI:** 10.3390/microorganisms13040710

**Published:** 2025-03-21

**Authors:** Ashwag J. Alzahrani, Basma M. Al-Hebshi, Zolfekar A. Yahia, Effat A. Al-Judaibi, Khloud H. Alsaadi, Awatif A. Al-Judaibi

**Affiliations:** 1Department of Biological Sciences, Microbiology Section, College of Science, University of Jeddah, Jeddah 21959, Saudi Arabia; ajaalzahrani1@uj.edu.sa (A.J.A.); basooma1@windowslive.com (B.M.A.-H.); eaaljedeibi@uj.edu.sa (E.A.A.-J.); khalsediy@uj.edu.sa (K.H.A.); 2Department of Internal Medicine, Al Noor Specialist Hospital, Ministry of Health, Makkah 24242, Saudi Arabia; zolfekaraliyahia@gmail.com

**Keywords:** inflammatory bowel disease, Crohn’s disease, ulcerative colitis, microbial diversity, intestinal microbiota, bacteria

## Abstract

Inflammatory bowel disease (IBD) is a chronic condition that includes two main types, Crohn’s disease (CD) and ulcerative colitis (UC), involving inflammation of the gastrointestinal (GI) tract. The exact cause of IBD is unknown but could be a combination of genetic, environmental, and immune system factors. This study investigated the impact of IBD on microbiota diversity by evaluating the differences in microbial composition and the microbiota of a control group (A) of healthy individuals and a group (B) of IBD patients. Sixty biopsies were collected from participants recruited from hospitals in Makkah, Saudi Arabia. Biopsy specimens were taken during colonoscopy examination, and bacterial identification was performed by extracting ribosomal DNA from sigmoid colon biopsies using a DNeasy Blood & Tissue Kit. Metagenomics and bioinformatics analyses were then conducted to analyze and compare the microbiota in the two groups. The results showed that the varieties of core microbiome species were 3.81% greater in the IBD patients than in the members of the control group. Furthermore, the differences between the groups were significantly greater than the variations within each group. Differences between the two groups were detected in the relative abundance of *Clostridium nexile*, *Ruminococcus gnavus*, *Ruminococcus faecis*, and *Escherichia coli*. These results indicate that microbiota could play a role in the pathogenesis of IBD and suggest that microbial diversity can serve as a biomarker for diagnosing the disease and monitoring its progression.

## 1. Introduction

The human microbiome refers to the collective genetic content of microorganisms residing in various parts of the human body, including the mucous membranes, skin, respiratory system, urinary and reproductive systems, and mammary glands [1]. Among these, the digestive system plays a particularly critical role in human health, functioning as an essential immune system [2]. The gut is home to trillions of microorganisms, collectively known as the microbiota, which interact with the host to influence digestion, metabolism, nutrient absorption, and waste excretion [2]. These microorganisms—including bacteria from major phyla such as *Actinobacteria*, *Proteobacteria*, *Firmicutes*, and *Bacteroidetes*—exist in a delicate balance that can be symbiotic, mutualistic, or pathogenic [1,3].

The diversity and composition of the gut microbiota are influenced by numerous factors, including inherited genetic traits, individual physiological conditions, diet, medication use, living environment, age, and hormonal changes [1]. Disruptions to the gut microbiome, referred to as dysbiosis, can have significant health consequences, including the development of inflammatory bowel disease (IBD) and antibiotic-resistant bacterial infections [4]. In the context of IBD, the melanocortin system has gained attention for its potential anti-inflammatory effects. Research has identified that melanocortin receptors, particularly MC1R, MC3R, MC4R, and MC5R, are expressed on immune cells such as macrophages, dendritic cells, and T cells [5]. Activation of these receptors has been shown to suppress the production of pro-inflammatory cytokines such as TNF-α, IL-6, and IL-1β, enhance the release of anti-inflammatory cytokines like IL-10, and regulate epithelial barrier integrity, which is often compromised in IBD patients [3,6]. The interplay between the gut microbiota and the melanocortin system offers novel insights into metabolic disorders and paves the way for innovative therapeutic strategies.

IBD is a chronic inflammatory condition that affects the gastrointestinal (GI) tract and encompasses two main types: Crohn’s disease (CD) and ulcerative colitis (UC). The etiology of IBD is complex, involving interactions between the gut microbiota, host genetics, and environmental factors [6,7].

Symptoms include GI bleeding, weight loss, malnutrition, and abdominal pain, which can progress from mild to severe over time [8].

UC is a long-lasting, immune-mediated inflammatory disorder primarily affecting the large intestines, with inflammation typically beginning in the rectum and potentially spreading to other sections of the colon; while UC can occur at any age, its onset predominantly peaks between 15 and 30 years of age [9]. Initial symptoms include rectal bleeding, urgency, and tenesmus (a sensation of incomplete evacuation). Although the disease has a low mortality rate, it causes significant morbidity, impacting social interaction, employment, and overall quality of life [10,11]. UC is also associated with psychiatric conditions such as anxiety and depression [12]. Over time, chronic inflammation increases the risk of dysplasia and colorectal cancer [13,14].

The incidence of UC is rising globally, affecting nearly one million individuals in both the United States and Europe [15,16]. Advances in treatment options have improved disease management, yet the increasing prevalence highlights the need for deeper insights into the disease’s underlying mechanisms and effective therapies.

Research on microbiota has revealed that certain microorganisms play beneficial roles in enhancing digestion, immunity, and metabolic processes, while others contribute to disease progression. Individuals with IBD often exhibit altered gut microbiota compared with healthy individuals, with a higher prevalence of certain bacterial species such as *Bacteroides* [3,17]. However, the relationship between immune response, bacterial diversity, and genetic predisposition remains unclear. Environmental factors, including dietary habits, antibiotic use, pollution, and overall health status, are believed to influence the development and progression of IBD [4]. Despite extensive research, the chronic and relapsing nature of IBD and its underlying mechanisms remain incompletely understood. Additional studies are required to elucidate the role of microbiota diversity in IBD pathogenesis and to identify effective strategies for prevention, early diagnosis, and treatment [18].

On the other hand, the intestinal microbiota plays a critical role in maintaining gut homeostasis, influencing immune regulation, nutrient metabolism, and protection against pathogenic invasion [19,20]. In IBD characterized by chronic inflammation of the gastrointestinal tract, microbial dysbiosis is a key contributing factor to disease onset and progression [21]. Dysbiosis in IBD is marked by a loss of microbial diversity, an increased presence of pro-inflammatory bacteria such as *Clostridium difficile*, *Escherichia coli*, *Campylobacter*, and *Fusobacterium*, and a reduction in beneficial commensals like *Faecalibacterium prausnitzii* and *Bacteroides* [19,22]. This altered microbial composition disrupts the intestinal barrier, triggers aberrant immune responses, and fuels chronic inflammation [20,23]. Furthermore, the microbiota does not act alone but interacts with host physiological systems, including the neuroendocrine pathways that regulate inflammation. The melanocortin system has recently gained attention for its role in immune modulation and gut homeostasis [24,25,26]. Therefore, this research aimed to isolate and identify intestinal microbiota using 16S rRNA in order to compare healthy controls (HCs) with patients with IBD, identify the abundance of 20 phyla in HCs and patients with IBD, and compare the formation of intestinal microbiota between HCs and patients with IBD.

## 2. Materials and Methods

### 2.1. Ethical Approval

The Local Committee for Research Ethics in the Makkah Region and the Health Directorate General of Health Affairs of the Makkah Region approved the study protocols (approval number H-02-K-076-1221-618; date of issue: 30 January 2022). All participants provided informed consent to participate in this study.

### 2.2. Sample Collection

A total of 60 sigmoid colon biopsies were collected from participants suffering from severe diarrhea with persistent bleeding, accompanied by intense abdominal pain and severe cramps, in addition to weight loss (19 patients with IBD and 41 HCs). All participants were adults (>18 years old); 33 were women and 27 were men. Biopsy specimens were taken during colonoscopy examination and between 2021 and 2023. Sampling included active and remission stages for each participant undergoing clinical examination, which included laboratory tests for elevated C-reactive protein, erythrocyte sedimentation rate, and exclusion of infections by fecal test, endoscopic examination, and histopathology examination, and the information of each participant was obtained from their medical records. All the samples were immediately stored at −80 °C before processing.

### 2.3. DNA Extraction and Sequencing Library Preparation

16S rRNA was isolated from the 60 intestinal biopsy samples for bacterial identification using the DNeasy Blood & Tissue Kit (Qiagen, Hilden, Germany) according to the manufacturer’s instructions [27].

For genomic DNA isolation, biopsy specimens (2 to 5 mg) were suspended in 180 µL of ATL buffer (a tissue lysis buffer) (Qiagen, Hilden, Germany), and 20 µL of proteinase K (at a concentration of 20 mg/mL) was added and thoroughly mixed by vortexing. The mixture was then incubated at 56 °C until the tissue was completely lysed (approximately 3 h). The sample was mixed by vortexing for 15 s, and 200 µL of buffer was added and mixed in the same manner. Then, 200 µL of ethanol (96–100%) was mixed in by vortexing. The mixture was pipetted into a DNeasy Mini spin column (Qiagen, Hilden, Germany), placed in a 2 mL collection tube, and centrifuged at 6000 rpm for 1 min. The DNeasy Mini spin column was placed in a new 2 mL collection tube, 500 µL of buffer AW1 was added, and the mixture was centrifuged for 1 min at 8000 rpm. Then, the DNeasy Mini spin column was placed in a new 2 mL collection tube, 500 µL of buffer AW2 was added, and the mixture was centrifuged for 3 min at 14,000 rpm. The DNeasy Mini spin column was then placed in a clean 2 mL micro-centrifuge tube, and 200 µL buffer AE was directly pipetted onto the DNeasy membrane, which was then incubated at room temperature for 1 min before centrifuging for 1 min at 8000 rpm to elute the DNA in a concentration of 80 ng/µL (16 µg). The isolated DNA samples were stored at −20 °C according to the manufacturer’s protocol, as outlined in the DNeasy Blood and Tissue Handbook [27,28]. DNA extractions were performed in the histopathology laboratory at King Fahad Medical Centre for Research, King Abdulaziz University.

For agarose gel electrophoresis, the extracted DNA samples were mixed with 6× DNA loading buffer (5:1 ratio) (Qiagen, Hilden, Germany) and placed into the wells. A 3 μL volume of DNA ladder was placed into wells alongside the gel. The horizontal electrophoresis (SCIEPLAS, Scie-Plas Ltd., Cambridge, UK) lead was firmly attached and connected to the power supply (Thermo ECEC135-90 electrophoresis Power Supply, Thermo Electron, Thermo Fisher Scientific Inc., Carlsbad, CA, USA), ensuring that the DNA migrated from the negative to the positive terminal. Electrophoresis was carried out at a voltage of 130 V and 100 mA for 1 h and 14–20 min. The migration distance of the DNA in the gel was judged by visually monitoring the migration of the tracking dyes. The DNA bands were visualized under UV light. The ethidium bromide (fluorescent dye) intercalated between the bases of DNA, allowing visualization of the bands. The 16S gene was PCR-amplified using the forward and reverse primers 518F: 5′-CCAGCAGCCGCGGTAATACG-3′ 800R: 5′-TACCAGGGTATCTAATCC-3′; 27F: 5′-AGAGTTTGATCMTGGCTCAG-3′; and 1492R: 5′-GGTTACCTTGTTACGACTT-3′. The primers for the amplification of the 16S gene were designed to target the conserved regions in the 518F, 800R, 27F, and 1492R genome sequences [29]. The sequenced data were identified using BLASTNCBI (v2.14.0) (http://blast.ncbi.nlm.nih.gov/Blast.cgi) (accessed on 21 March 2023), and the sequences were analyzed by BGI Genomics (Hong Kong, China; order number F22FTSAPHT1367-01) (MyBGI 3.0) (accessed on 21 March 2023).

### 2.4. Metagenomics Analysis

A total of 30 ng of qualified DNA template and the 16S rRNA fusion primers were added for the polymerase chain reaction (PCR). All PCR products were purified with Agencourt AM Pure XP beads (Beckman Coulter, Indianapolis, IN, USA), dissolved in elution buffer, and eventually labeled to finish the library construction. Library size and concentration were detected using an Agilent 2100 Bioanalyzer (Agilent, Santa Clara, CA, USA). Qualified libraries were sequenced according to their insert size.

### 2.5. Bioinformatics Analysis

Raw data were filtered to obtain high-quality data, after which clean reads that overlapped were merged with tags and further clustered into operational taxonomic units (OTUs). To display the number of common and unique OTUs of HCs and participants with IBD, we constructed a Venn diagram using R software (v3.1.1), while taxonomic classifications were assigned to an OTU representative sequence using the Ribosomal Database Project database and obtained with R software (v3.1.1). Alpha diversity, beta diversity, and differential species analyses and network and model predictions were carried out based on the OTU profile table and taxonomic annotation results using the following software: Fast-Tree (version 2.1.3 http://www.microbesonline.org/fasttree/ accessed on 23 March 2023), the “NMF v0.20.5” package in R software (v3.1.1), and GraPhlAn (v1.1.3) (https://huttenhower.sph.harvard.edu/graphlan/ accessed on 23 March 2023).

Unweighted pair group method with arithmetic mean (UPGMA) cluster analysis was carried out according to the weighted UniFrac and unweighted UniFrac distance matrix [30]. Clustering results were integrated with the relative abundance of species at different taxonomic levels to display the differences in species composition between samples.

### 2.6. Statistical Analysis

Statistical analyses were performed using the Statistical Package for the Social Sciences, version 20 (IBM, Armonk, NY, USA). Differences between samples and homogeneity between groups were determined using the chi-square test and correlation coefficient test. Results were considered significant at an alpha level of *p* ≤ 0.05 and highly significant at *p* ≤ 0.01.

The total number of bases, reads, GC (%), Q20 (%), and Q30 (%) were calculated for each isolated bacterium. Phylogenetic indices serve as a quantitative index that shows how many different species are found in a dataset (a community), as well as the phylogenetic relationships (co-distribution, species affinity, species richness) between distributed individuals. After sequencing, the FastQC (v.3.4.1) program was used for quality control steps. Following the quality control analysis, data quantities, read qualities, GC distributions, kmer distributions, and possible adapter contamination of each sample were determined and evaluated with mothur (v.1.31.2) [31]. Readings with poor reading quality (Phred score < Q20, 30 bp window range) were excluded from further analysis. Further, low-quality base reads at terminal regions and chimeric sequences with possible adapter contaminants were removed using the Trimmomatic tool and Genomes OnLine Database (GOLD) (v.0.40) [32]. Taxonomic profiling was performed using Kraken2 v2.1.2 [33], and the SILVA database v 138 (2020) was used as a reference dataset [34]. OTU groups in each sample were determined after alignment. R scripts were used for data reporting, statistical analysis, and data visualization (https://www.R-project.org/ accessed on 23 March 2023) R (v.3.4.1).

Alpha diversity refers to the analysis of species diversity in a sample and is measured using the observed species index, Chao index, ACE index, Shannon index, Simpson index, and Good’s coverage index. Species diversity is proportional to the first four values, whereas the Simpson index is negatively correlated with species diversity. Additionally, a higher Good’s coverage value indicates fewer undiscovered species in the given samples. The observed species index, Chao index, and ACE index reflect the species richness in the microbial community, and rarefaction curves of these three indices can be drawn to evaluate whether sequenced data are sufficient to cover all potential species. Sequenced data are thought to be sufficient when the rarefaction curves are smooth; otherwise, increasing the sample size is recommended. The Shannon index and Simpson index reflect the species diversity of the community, which is affected by species richness and species evenness. Given the same species richness, greater species evenness indicates greater species diversity. The Good’s coverage index shows the coverage of the sample library. A smaller index value indicates a lower probability that sample sequences are not sequenced. This value indicates whether the sequencing results can represent the real property of the sample [35,36,37].

## 3. Results

This study investigated the microbiota composition in HCs and IBD patients. The clinical and demographic data of participants showed that 42.11% (of 19) of IBD patients were women, while 57.89% were men. While the percentage of IBD patients who were between the ages of 20 and 30 years was 0%, the percentage of the IBD patients who were between the ages of 31 and 40 years was 13.33%, and this percentage decreased in IBD patients with ages between 41 and 50 years to become 8.33%; furthermore, IBD patients with ages between 51 and 60 years reached a percentage of 8.33%, and those with ages more than 60 years reached a percentage of 1.67%, compared with 12.73%, 9.09%, 9.09%, 27.27%, and 10.16% of control participants in categories of ages 20–30, 31–40, 41–50, 51–60, and >60, respectively. On the other hand, the prior family history of IBD showed that 17.74% of patients had a prior family history of IBD, and 16.39% were negative for prior IBD family history, compared with 31.15% and 34.72% of control participants who had positive and negative prior family histories, respectively. The chi-square statistical result showed that there is a significant relationship between having a family history of the disease and the presence of IBD (*p*-value = 0.047). This implies that individuals with a family history of IBD are more likely to have the disease themselves. This was confirmed by the result of correlation coefficient analysis (*p*-value = 0.00) (Appendix A).

### 3.1. OTU Analysis

OTU clustering was carried out for the sequences at 97% similarity, and a total of 2353 OTUs were obtained, with an average of 28 OTUs in the control group and 63 OTUs in the IBD group. The Venn diagram of OTUs showed that the healthy control group had 1150 OTUs, the IBD group had 1203 OTUs, and the two groups shared 961 OTUs. The proportion of unique core species in the microbiomes of the IBD patients was 17.39%, and the proportion in the control group was 13.58%. Most of the species (69.03%) were common to both groups. The varieties of core microbiome species in IBD patients increased by 3.81% when compared with the healthy controls (Figure 1a).

The rank charts show two aspects of species diversity: species abundance and species evenness. In the horizontal direction, the width of the curve reflects the abundance of the species. The more the curve crosses the horizontal axis, the higher the abundance of a species. The smoothness of the curve reflects the species uniformity in the sample. The smoother the curve, the more homogeneous the species distribution; the steeper the curve, the more heterogeneous the species distribution. The transversal range of the IBD group was larger than that of the control group, indicating increased species richness. The curve was steep, indicating decreased species evenness (Figure 1b).

### 3.2. Diversity Analysis

This section reports on the taxonomic diversity of the bacterial microbiota isolated from biopsies of healthy control participants and IBD patients.

#### 3.2.1. Alpha Diversity Analysis

The statistical results in Figure 2 show the analysis of alpha diversity and richness in the two groups, which were measured using the observed species, Chao, exponent, Shannon, Simpson, and Good’s coverage indices. The observed species diversity and Chao indices were lower in the HC participants (group A) than in the IBD patients (group B), while the exponent index indicated that group B had greater diversity. The Shannon and Simpson indices, which are negatively correlated with species diversity, were represented by small percentages in the two groups, while the percentages of Good’s coverage index were similar for both groups in terms of species richness. Figure 2a shows the observed species index results, which indicate that species richness was greater in group B than in group A, with a *p*-value of 0.181. The Chao index results indicate greater species richness in group B than in group A, with a *p*-value of 0.32 (Figure 2b). The ACE index results indicate greater species richness in group A than in group B, with a *p*-value of 0.6 (Figure 2c). The Shannon and Simpson indices reflect the species diversity of a community, which is affected by species richness and species evenness. Given the same species richness, greater species evenness indicates greater species diversity. The Shannon index score for group B was higher than that for group A, with a *p*-value of 0.372, while the Simpson index was higher for group A than group B, with a *p*-value of 0.683 (Figure 2d,e). The results of the Good’s coverage index indicated a lower probability of unsequenced samples in group A than in group B, with a *p*-value of 0.545 (Figure 2f).

#### 3.2.2. Beta Diversity Analysis

Beta diversity analysis, illustrated using a heatmap, allows us to evaluate differences in species complexity in a sample. Bray–Curtis values, which are a common indicator of the differences between two communities, were used for this purpose. The values range from zero to one, with zero representing a completely similar community structure.

As shown in Figure 3a, beta diversity analysis revealed differences in microbiota communities between the control participants (A) and IBD patients (B). The clustering patterns show that the microbiota of the control participants is more homogeneous, indicating lower beta diversity within this group. In contrast, the IBD patients exhibit higher beta diversity, with more heterogeneous microbial communities. This suggests that IBD significantly disrupts the gut microbiome, leading to greater variability and distinct compositional differences compared with HCs. The clear separation between the two groups underscores the impact of IBD on gut microbiota composition. Nonetheless, some similarities can be observed: despite the overall higher beta diversity within the IBD group, certain samples from both groups cluster, indicating shared microbial community characteristics.

The non-metric multidimensional scale (NMDS) results for OTU are presented in Figure 3b and indicated that six of the samples from the IBD group had features different from those in the samples from the HCs. The results identified possible factors in these six samples from the IBD patients. These differences can be interpreted in combination with the sample feature information; these may include gender, age, smoking (Appendix A), or the type of IBD (CD or UC). In our results, six of the IBD patients were over 51 years of age.

The samples were clustered, and the distances between them were calculated to judge the similarity in the composition of the species in each sample. As shown in Figure 3c, the microbiota species composition of six IBD patients showed dissimilarity with the microbiota of the control and the other IBD patients; however, they showed a degree of dissimilarity between them. The results showed a significant difference in beta diversity between the two groups (*p*-value = 0.012).

The results of the UPGMA analysis were integrated with the relative abundance of the species at various taxonomic levels to reveal the differences between the species. Figure 3c shows the results of the UPGMA cluster tree, and Figure 3d shows a histogram of species abundance. A shorter branch length between samples indicates greater similarity in the microbiota species composition, while a longer distance indicates greater differences between the control (A) and IBD (B) groups. The results showed a more diverse species composition in the samples from group B compared with the samples from group A. In terms of species composition, the highest species composition values were for *Actinobacteria*, *Alphaproteobacteria*, *Clostridia*, and *Bacteroidia*, while the lowest species composition values were for *Bacilli*, *Fusobacteria*, *Erysipelotrichia*, *Negativicutes*, and others (Figure 3d).

### 3.3. Taxonomic Profile

#### 3.3.1. Structural Analysis of Microbiota Isolated from the Biopsies of Control Participants and IBD Patients

Figure 4 shows the composition and proportions of microbiota in group A and B samples at the class level. Species were classified as “others” when their relative abundance was less than 0.5%. The composition and abundance of species were analyzed in samples isolated from the biopsies of the control participants (A) and the IBD patients (B). The abundance of *Alphaproteobacteria* species was low in group B, and these species did not appear in group A, while *Fusobacteria*, *Negativicutes*, and *Gammaproteobacteria* species were equally abundant in the two groups, and *Erysipelotrichia* species were rare in both groups. *Betaproteobacteria*, *Actinobacteria*, and *Bacilli* species were more abundant in group B than in group A. In contrast, *Bacteroidia* and *Clostridia* species were more abundant in group A than in group B (Figure 4a).

Furthermore, the abundance of microbial species can be analyzed based on color intensity and differences, with red representing the most abundant microbe, *Clostridia*. The abundance gradually decreased, reflected in the lighter color of *Gammaproteobacteria* and *Bacteroidia* microbes. The abundance of these microbes was similar in groups A and B. In contrast, *Actinobacteria*, *Betaproteobacteria*, and *Bacilli* microbes were more abundant in group B than in group A, and *Negativicutes* microbes were not abundant in either group. *Cytophagia*, *Flavobacteriia*, *Deinococci*, *Cyanobacteria*, and *Verrucomicrobiae* microbes were not abundant in group B or A. *Alphaproteobacteria* microbes appeared to be more abundant in group B than in group A, *Fusobacteriia* microbes appeared to be more abundant in group A than in group B, and *Deltaproteobacteria* and *Erysipelotrichia* microbes appeared to be equally abundant in the two groups (Figure 4b).

#### 3.3.2. Differential Phylum and Class Screening (Wilcoxon Test) and Comparison of Key Phyla and Classes

Figure 5a shows the results of the Wilcoxon test for the differential phylum screening, which was applied to predicted functions to assess the significance of the differences between the microbiome samples isolated from the control participants and the IBD patients. The percentage of relative abundance of Proteobacteria was 15% in the IBD patients and 10% in the control participants, showing a difference of 5%. There was a difference of 1% in the relative abundance of the phylum Actinobacteria in the IBD patients compared with the control participants. On the other hand, *Firmicutes* had the highest relative abundance in both groups, with an increase of 20% in the control participants compared with the IBD patients; for *Bacteroides*, there was an increase of 7% in the control participants compared with the IBD patients. Furthermore, the Log^2^(A/B) results show that *Tenricutes*, *Verrucomicrobia*, and *Candidatus Saccharibacteria* were detected in IBD patients. The results at the phylum level showed a significant difference between the two groups (*p*-value and FDR ≥ 0.05). The differential comparison of the phyla identified in the control participants and IBD patients showed that the presence of *Candidatus Saccharibacteria*, *Spirochaetes*, and *Tenricutes* in the IBD patients significantly differed from the control participants (Figure 5b).

The results of the Wilcoxon test for differential class screening are shown in Figure 6a. The control participants showed increased relative abundance of *Bacteroidia* and *Clostridia* compared with the IBD patients, with percentages of 20% and 7%, respectively, while the relative abundance of *Betaproteobacteria* and *Actinobacteria* was 1% higher in the IBD patients than in the control participants. Moreover, the Log^2^(A/B) results showed that *Acidobacteria GP16*, *Sphingobacteria*, *Mollicutes*, *Alphaproteobacteria*, and *Verrucomicrobiae* were detected in the IBD patients but not in the control participants. The results for classes showed a significant difference between the two groups (*p*-value and FDR ≥ 0.05). The differential comparison of the classes between the control participants and the IBD patients showed differences in the relative abundance of *Betaproteobacteria*, *Bacilli*, *Actinobacteria*, and *Gammaproteobacteria*. The results for *Betaproteobacteria* and *Bacilli* showed significant differences between the two groups, while *Alphaproteobacteria* showed a large significant difference in relative abundance (Figure 6b).

#### 3.3.3. GraPhlan Map

The results in Figure 7 show the GraPhlan map. From the diagram, it can be seen that healthy individuals (A) tend to have a more diverse microbiome with higher representation across multiple taxa, particularly *Firmicutes* and *Bacteroidetes*, which are essential for a balanced gut microbiota. In contrast, the IBD group (B) shows reduced diversity, especially in beneficial taxa such as *Firmicutes*, which are often associated with anti-inflammatory effects. Furthermore, the most increased taxon is *Clostridia*, followed by *Lachnospiraceae*, *Faecalibacterium*, and *Ruminococcaceae*. The red color shows an increase in *Proteobacteria*, specifically *Enterobacteriaceae*, in the IBD patients; this phylum is associated with pro-inflammatory responses. This is followed by *Gammaproteobacteria*, *Enterobacteriales*, *Enterobacteriaceae*, and *Escherichia*. The green color shows that Bacteroidetes is one of the primary phyla in the gut microbiome, associated with healthy digestion and metabolic functions. In the context of this study, it appears that the microbiota of both healthy individuals (A) and IBD patients (B) contain members of this phylum, although the relative abundance and diversity within specific genera could vary between groups. This is followed by *Bacteroidaceae*, *Bacteroides*, and *Actinobacteria*, which is the least phylogenetically advanced group.

#### 3.3.4. Genus of Bacterial Community

Figure 8 shows the relative abundance of the genera of microbes in all participants and between control participants (group A) and IBD patients (group B). Clustering of the samples based on the genera’s abundance in all participants was performed to determine the degree of similarity. Similarities were detected in the genera *Eubacterium*, *Ruminococcus*, *Lactobacillus*, and *Micrococcus*. Differences between the two groups were detected in *Ruminococcus*, *Corynebacterium*, *Escherichia*, *Clostridium_XVIII*, and *Bacteroides*, with higher relative abundance in IBD groups (Figure 8a). On the other hand, the clustering of the samples in the two groups showed similarities in the abundance of *Prevotella* and *Lachnospiracea_*incertae_sedis and differences in the abundance of *Pseudomonas*, *Prevotella*, and *Clostridium_XIVa* (Figure 8b). The relative abundances of phylum, class, order, family, genera, and species are shown in Appendix A.

### 3.4. Discriminant Analysis

Linear discriminant analysis (LDA) of linear discriminant analysis effect size (LEfSe) is used to identify high-dimensional biomarkers and reveal genomic characteristics concerning genes, metabolism, and classification. IBD changes the taxonomic biomarkers in each group. In addition, to identify the characteristic bacteria specific to each group, an LDA of LEfSe algorithm approach was applied.

In Figure 9, the various colors serve to distinguish the groups. The node colors represent vital biomarkers in the group with the same color as the nodes. The biomarker legends are shown in the top-right corner. The yellow nodes represent unimportant biomarkers in various groups. The six circles in Figure 9 represent, from inner to outer circle, the phylum, class, order, family, genus, and species levels. The microbiota isolated from the biopsies of the control participants and IBD patients included *Bacteroidales*, *Bacteroidia*, *Thermaceae*, *Thermales*, *Spirochaete*, *Spirochaetes*, and *Spiochaetia* (Figure 9a).

In an LDA diagram, the colors of the bars distinguish important microbial biomarkers in the groups, and the lengths of the bars indicate the LDA scores. In Figure 9b, red represents the control group (A) and green represents the IBD group (B). The comparison results show a difference in the lengths of the groups based on the degrees of the LDA. Taxa abundance analysis showed that 15 microbiota species were differentially abundant in the control group (LDA score > 1) and 97 microbiota species were differentially abundant (LDA > 1) in the IBD group (Figure 9b).

## 4. Discussion

In this study, we investigated and identified differences in the microbial composition of the biopsies obtained from healthy control participants and IBD patients; we used this method because it provides a direct and precise representation of the microbial communities at the mucosal level. Unlike stool samples, which mainly reflect luminal microbiota, biopsies express bacteria in close association with the intestinal epithelium, where critical host–microbe interactions occur. This is particularly important in IBD, as dysbiosis and microbial translocation at the mucosal interface contribute to inflammation and disease progression. Several studies have highlighted the importance of sampling methods in analyzing the gut microbiota of individuals with inflammatory bowel disease (IBD), and biopsy samples found significant differences in microbial composition among these sample types [38,39,40]. The latter act as a model of a system that causes inflammation and tissue death due to microbial dysbiosis and an abnormal immune response [41]. Recently, there has been increased awareness of the gut microbiome’s role in IBD [42], and in general, the IBD patient’s gut microbiome is characterized by reduced diversity, a decline in anti-inflammatory bacteria, and an increase in pro-inflammatory bacteria [43]. The percentage of unique varieties of core microbiome species in IBD patients was 17.39%, and their percentage in the control participants was 13.58%; the variety of core microbiome species in the IBD patients suggests that it is 3.81% higher than those in the healthy control group. These results may be due to differences in study design, small sample size (60 cases), and an inability to control for risk factors such as smoking [44]. Our findings agree with earlier research that found that bacterial infections are more prevalent in IBD patients than in individuals without IBD [45]. Furthermore, species richness rose in the IBD group, but species evenness reduced in the control group. We observed an increase in alpha diversity, as measured by the Shannon index, in the IBD group. However, this finding was not corroborated by the Simpson index, which is considered more reliable for small sample sizes. Additionally, the Chao index results suggest greater species richness in group B compared with group A, while the ACE index indicates the contrary. However, there was no significant difference in the alpha diversity between biopsies of the healthy control participants and IBD patients. The discrepancy between the Shannon, Simpson, Chao, and ACE indices can be attributed to the different approaches each index uses to measure community diversity, e.g., species richness and evenness. Consequently, the indices frequently produce contrasting results due to differences in computational focus [43], particularly in small sample sizes or when certain species dominate the community structure. In our study, significant variations in the diversity index values can result from relatively small variations in sample size such as gender (proportion of males and females), age, or disparate medical histories (data from patients’ medical records are available in Appendix A). Our data are consistent with the findings of [44], which demonstrated that advancing age and certain drugs might alter the microbiota, potentially leading to inflammatory processes. Additionally, there were variations in alpha and beta diversity between healthy individuals and IBD patients who were in remission.

When analyzing beta diversity, it was found that there was a significant difference in community composition between the two groups. Furthermore, the microbiota species composition of six IBD patients showed dissimilarity compared with the microbiota of the control and the other IBD patients; this can be interpreted based on the features of these samples. Furthermore, the microbiota species composition of six IBD patients showed a difference compared with the microbiota of the control and the other IBD patients; this can be interpreted based on the features of these samples, which may be influenced by a variety of factors. Interestingly, our findings suggest that these six samples may correspond to IBD patients aged over 51 years. This finding underscores the importance of considering age as a significant factor influencing gut microbiota composition in IBD patients. Previous studies have indicated that age can significantly influence gut microbiota composition, with older individuals showing distinct microbial patterns [45,46]. This approach could lead to more precise and effective treatments, improving the quality of life for older individuals with IBD. On the other hand, there is a significant relationship between having a family history of the disease and the presence of IBD (*p*-value = 0.047). This was confirmed by the results of correlation coefficient analysis (*p*-value = 0.0001) (data from patients’ medical records are available in Appendix A).

We investigated differences between the groups in terms of community composition. An analysis of the metagenome showed that the abundance of *Betaproteobacteria*, *Actinobacteria*, and *Bacilli* in the IBD group was significantly higher than in the control group. Furthermore, several researchers have mentioned a possible link between IBD and gut microbiota [47]. According to Pittayanon et al. [48], patients with UC have reduced diversity, lower proportions of *Bacteroides* and *Faecalibacterium*, and higher numbers of *Betaproteobacterium*. Additionally, the abundance of *Bacteroidia* and *Clostridia* species was significantly higher in the control group compared with the IBD group since the major species in balanced gut microbiota are *Bacteroidia* and *Clostridia* [49]. Bacteroidetes and other helpful bacteria support immunological control, nutrient absorption, and digestion, all of which contribute to gut homeostasis [50,51]. Gut microbiota changes, such as reduced Bacteroidetes, may disrupt these processes, leading to inflammation and other UC symptoms [52,53]. Consequently, IBD may be linked to a decline in beneficial taxa (i.e., Bacteroidia and Clostridia) and an increase in harmful taxa (e.g., Proteobacteria and Actinobacteria) in gut microbiota. The level of *Escherichia coli* (phylum Proteobacteria) was elevated only in CD patients [54]. This might indicate that *E. coli*, a bacterium that plays a major role in IBD development, is present, while *Actinobacteria* were more prevalent in UC patients [54]. A similar finding was obtained in a study of 61 UC patients and 61 healthy individuals [55]. It has been demonstrated that *actinobacteria* can modulate immune responses in the GI tract. UC is associated with abnormal immune responses to gut microbiota [48,56]. A change in the composition of *Actinobacteria* could affect the local immune response, increasing or reducing the level of inflammation [48,56].

The most prevalent phyla in both the control participants and IBD patients are *Firmicutes*, *Actinobacteria*, *Bacteroidetes*, *Verrucomicrobia*, and *Proteobacteria*; however, the diversity and richness of each of these groups vary according to the severity of the disease [57,58]. The loss of typical anaerobic bacteria is the primary reason for the reduced microbial diversity in IBD; *Bacteroidales*, *Faecalibaculum*, and some *Clostridiales*, including *Faecalibacterium prausnitzii*, *Bifidobacterium*, and *Lactobacillus*, are less common in IBD patients compared with Enterobacteriaceae such as *E. coli*, *Pasteurellaceae*, *Fusobacteriaceae*, *Neisseriaceae*, *Veillonellaceae*, *Gemellaceae*, *Ruminococcus*, and *Clostridium* spp., which are more prevalent in IBD patients than in healthy individuals [59,60,61].

The four major bacterial phyla—*Firmicutes*, *Proteobacteria*, *Actinobacteria*, and *Bacteroidetes*—as well as the species *Bifidobacterium adolescentis*, *Faecalibacterium prausnitzii*, *Bacteroides ovatus*, *Escherichia coli*, and *Ruminococcus gnavus* of the *Lachnospiraceae* family were the subjects of our study. We focused on these taxa because they all appear to play an intriguing role in the pathophysiology of IBD. The *Lachnospiraceae* family is primarily composed of anti-inflammatory butyrogenic species and is decreased in IBD patients, rising in proportion to the disease’s remission [62]. This family includes several genera of the *Clostridia* cluster, including XIVa, XIVb, IV, and *Faecalibacterium prausnitzii* [62]. Compared with HCs, the mucolytic bacterium *Ruminococcus gnavus* was elevated in IBD patients and is thought to be a potential biomarker of mucosal injury [63]. *Bifidobacterium adolescentis* has been shown to be attenuated in the tissue microbiome of IBD patients, based on examination of their tissue microbiome. Bifidobacteria are beneficial for maintaining intestinal barrier functions, as well as for producing short-chain fatty acids (SCFAs) [64]. Lower relative abundance of *Bacteroidetes* and higher abundance of *Proteobacteria* have been observed in high antibody titers by targeting the antigens of *Bacteroides ovatus*, a bacterium that seems to be involved in the pathophysiology of IBD [65]. This study has limitations, including the sample size, potential variability in biopsy site selection and preparation, the comparison between CD and UC, and the influence of external factors, such as age, smoking, gender, and diabetes, on microbiome composition.

## 5. Conclusions

The results of this study demonstrate that IBD is associated with significant microbial dysbiosis characterized by altered diversity, disrupted community structure, and taxonomic shifts in gut microbiota. Alpha diversity analysis revealed that IBD patients have higher microbial richness but lower evenness, indicating an imbalanced microbial community. Beta diversity analysis further confirmed that IBD significantly alters gut microbiota composition, leading to greater microbial heterogeneity compared with healthy controls.

At the taxonomic level, IBD patients showed increased abundance of pro-inflammatory taxa, including *Proteobacteria*, *Enterobacteriaceae*, *Escherichia*, *Clostridia*, and *Actinobacteria*, while beneficial taxa such as *Firmicutes*, *Lachnospiraceae*, *Faecalibacterium*, *Ruminococcaceae*, and *Bacteroidetes* were more abundant in healthy controls. The LEfSe biomarker analysis identified 97 significantly enriched taxa in IBD patients compared with only 15 in healthy individuals, with key microbial markers such as *Bacteroides*, *Thermaceae*, *Spirochaetes*, and *Enterobacteriaceae* being associated with IBD.

These findings suggest that IBD is characterized by a loss of microbial stability, an overrepresentation of potentially pathogenic bacteria, and a decline in beneficial, anti-inflammatory species, which may contribute to disease pathogenesis and immune dysregulation.

## Figures and Tables

**Figure 1 microorganisms-13-00710-f001:**
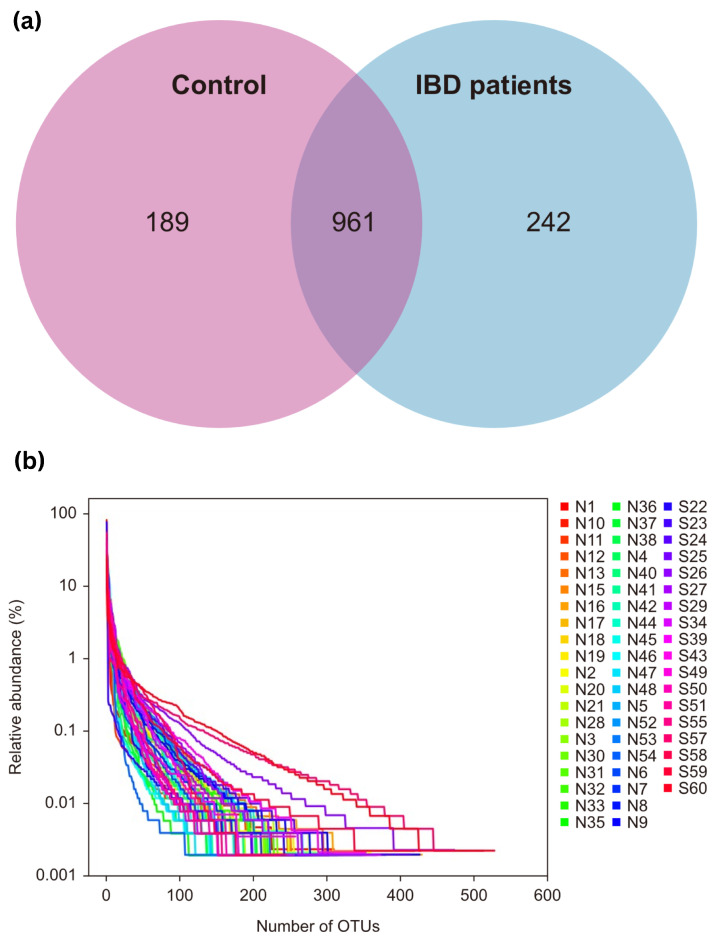
(**a**) Venn diagram of OTUs; (**b**) OTU rank curve between control participants (group A, “shown as N”) and IBD patients (group B, “shown as S”). OTUs—operational taxonomic units.

**Figure 2 microorganisms-13-00710-f002:**
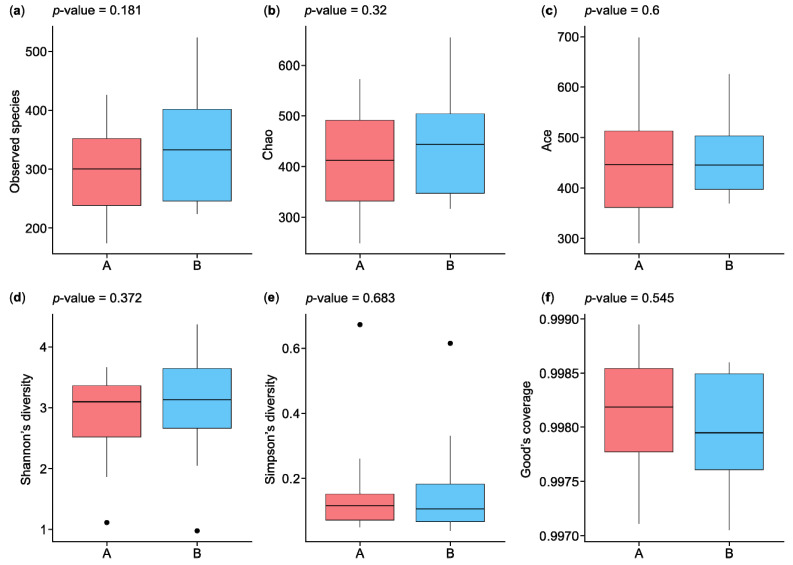
Boxplots of alpha diversity indices of microbiota species isolated from the biopsies of control participants (A) and IBD patients (B): (**a**) observed species, (**b**) Chao, (**c**) ACE, (**d**) Shannon’s diversity, (**e**) Simpson’s diversity, and (**f**) Good’s coverage. IBD—inflammatory bowel disease. ⦁ Represent outliers, which are data points that fall outside 1.5 times the interquartile range (IQR) from the first or third quartile.

**Figure 3 microorganisms-13-00710-f003:**
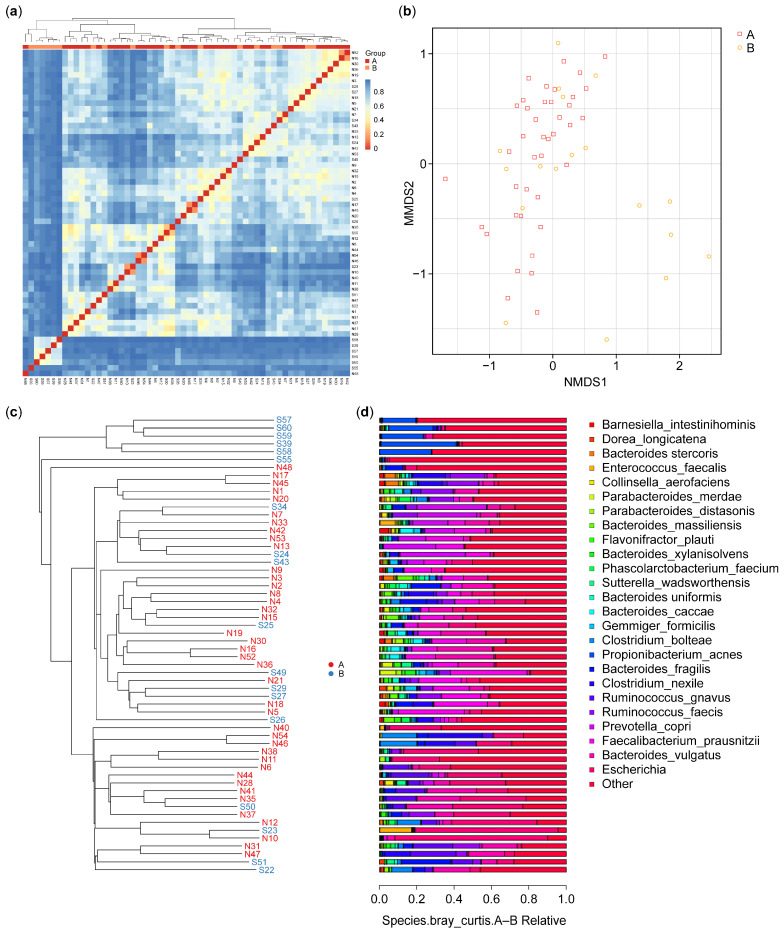
Beta diversity: (**a**) heatmap of the microbiota species isolated from the biopsies of the control (A) and IBD patient (B) groups; (**b**) OTU NMDS analysis between the control (A) and IBD patient (B) groups; (**c**) UPGMA cluster tree for control (N) and IBD patients (S); (**d**) species relative abundance bar plot of microbiota isolated from the biopsies of healthy control participants (A) and IBD patients (B), each bar corresponding to its front sample labeled in figure (**c**). This correspondence enables the identification and interpretation of the microbial composition for each sample based on the phylogenetic tree structure. IBD—inflammatory bowel disease; OTU—operational taxonomic unit; UPGMA—unweighted pair group method with arithmetic mean.

**Figure 4 microorganisms-13-00710-f004:**
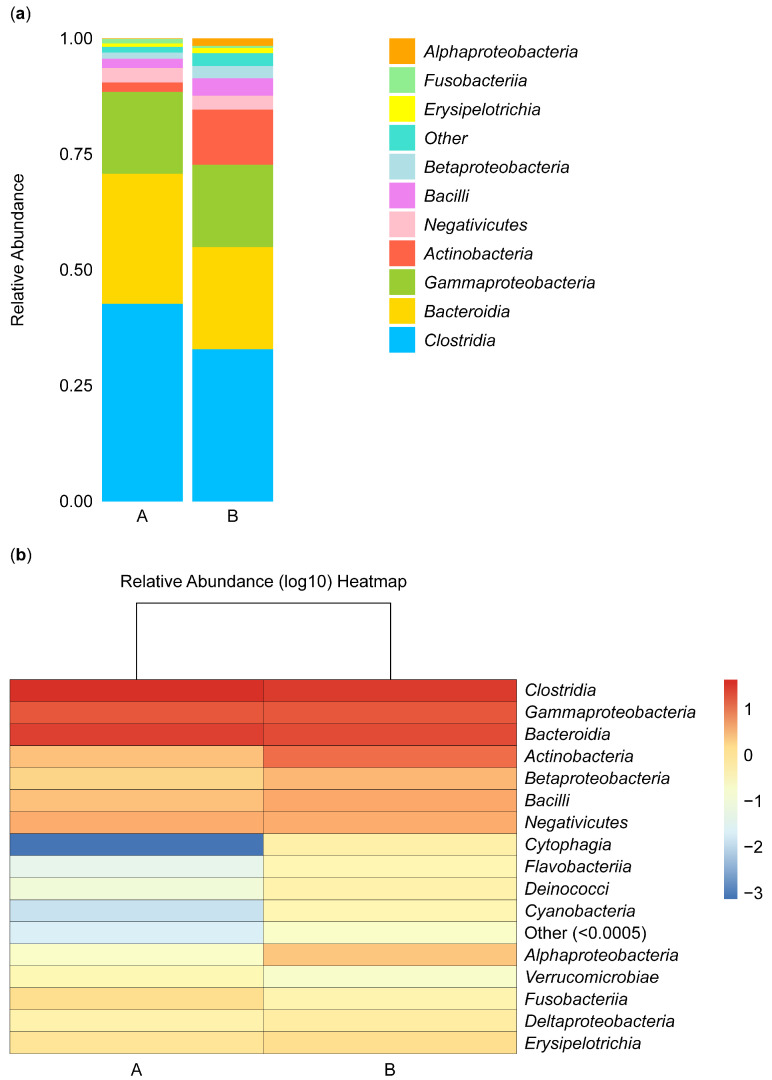
Stacked bar plot of class-level taxa (**a**) and heatmap (**b**). The stacked bar illustrates the relative abundance of the taxa identified for microbiota isolated from the biopsies of the control participants (A) and IBD patients (B) at the class level.

**Figure 5 microorganisms-13-00710-f005:**
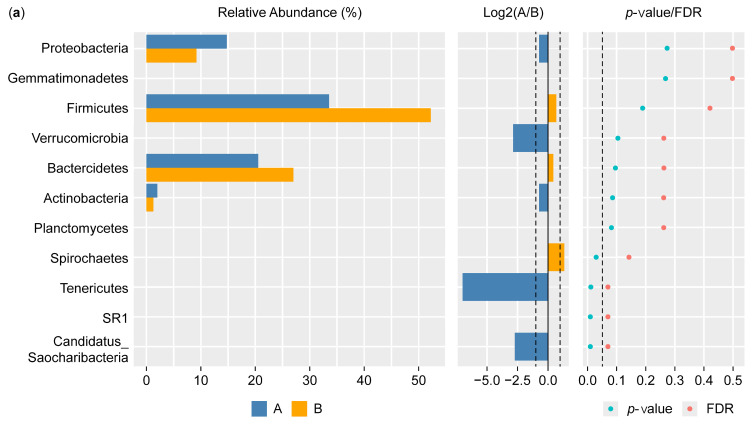
Differential phylum screening using the Wilcoxon test showing (**a**) key differential phyla and (**b**) relative abundances of 10 phyla in the microbiota isolated from the biopsies of the control participants (A) and IBD patients (B); the figure in the upper-right corner is a magnification of the invisible six phyla, illustrating the variation in phylum distribution between the two groups. * The differences between the groups are significant at *p* ≤ 0.05; ** the differences between the groups are highly significant at *p* ≤ 0.01. FDR: false discovery rate. If FDR values are less than 0.05, there is a significant difference between the two groups.

**Figure 6 microorganisms-13-00710-f006:**
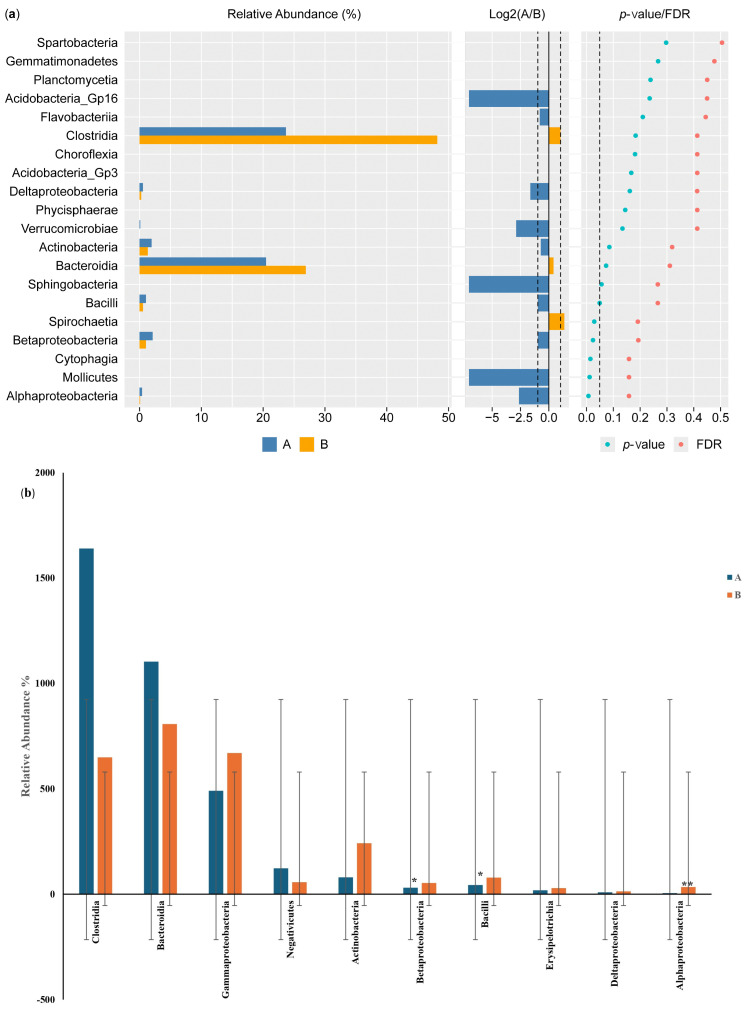
Differential class screening using the Wilcoxon test showing (**a**) key differential classes and (**b**) relative abundances of 10 classes in the microbiota isolated from the biopsies of the control participants (A) and IBD patients (B). * The differences between the groups are significant at *p* ≤ 0.05; ** the differences between the groups are highly significant at *p* ≤ 0.01. FDR: false discovery rate. If FDR values are less than 0.05, there is a significant difference between the two groups.

**Figure 7 microorganisms-13-00710-f007:**
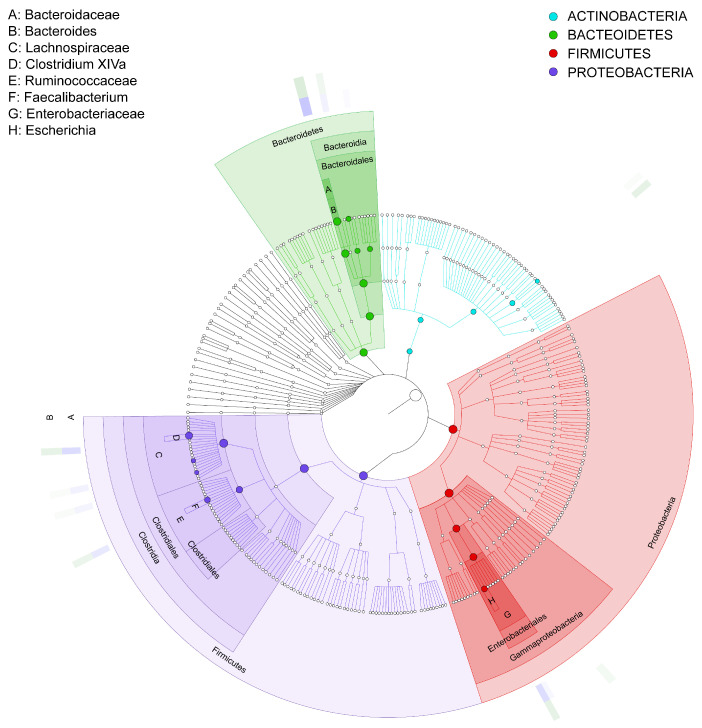
GraPhlan map of bacterial taxa based on the phyla of the microbiota communities isolated from biopsies of healthy control participants (A) and IBD patients (B). The labels A–H represent the bacterial phyla in the samples, and the bold circles illustrate the taxonomic gradient ranging from the phylum to the genus level. The color distribution highlights bacterial groups that show statistically significant differences between the two groups. Dark circles indicate different taxonomic levels, ranging from phylum to genus, demonstrating the hierarchical structure of bacterial classification. The color distribution further highlights bacterial groups that significantly differ between the two groups (IBD and healthy participants), as determined by statistical analysis.

**Figure 8 microorganisms-13-00710-f008:**
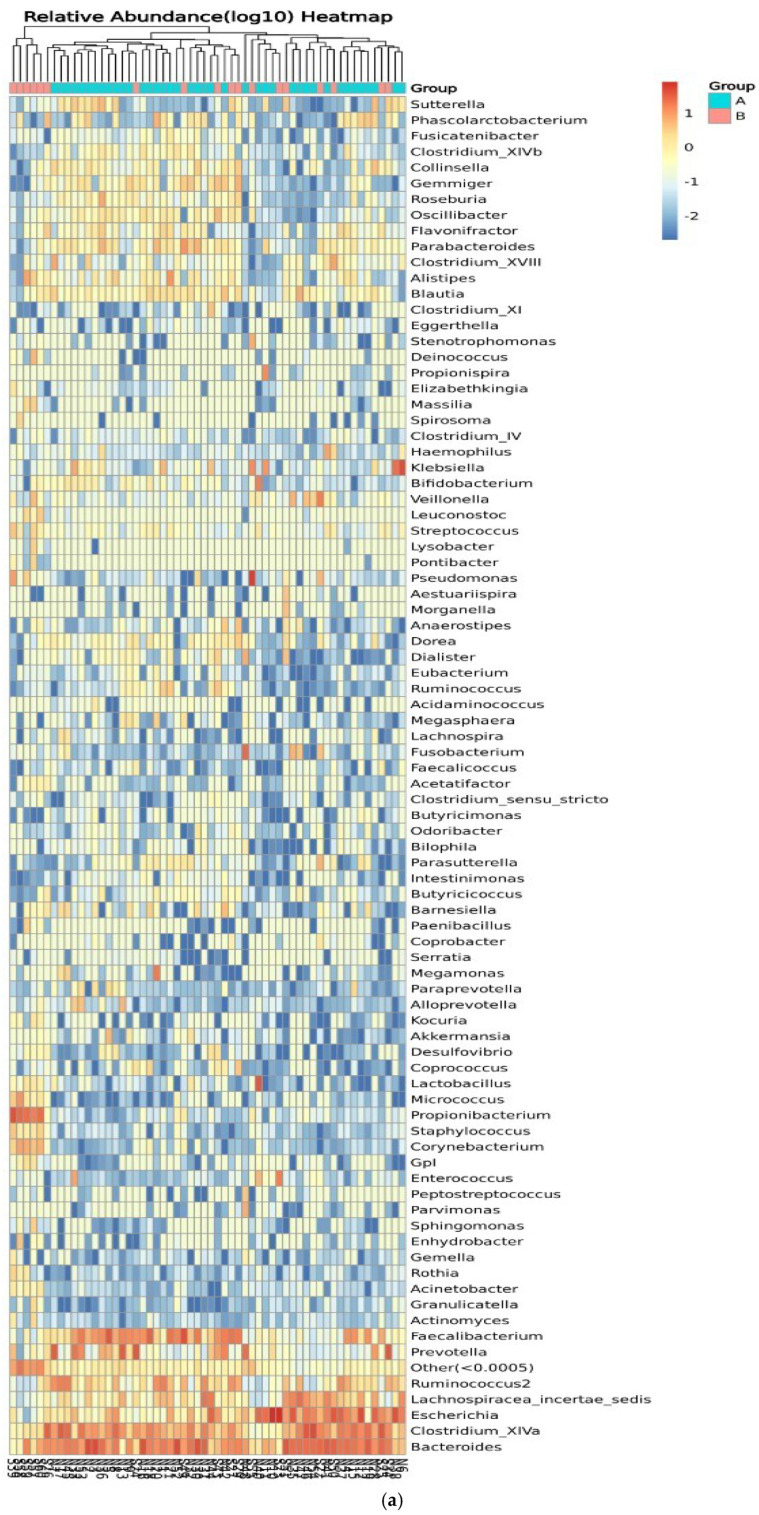
Heatmaps of the genera detected in the microbiota isolated from biopsies: (**a**) genera detected in each participant’s sample, and (**b**) comparison between control participants (A) and IBD patients (B).

**Figure 9 microorganisms-13-00710-f009:**
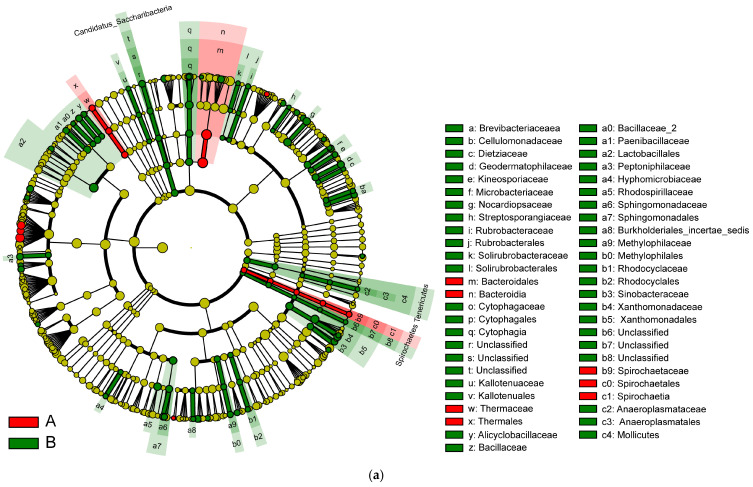
(**a**) LEfSe analysis cluster; (**b**) LDA of LEfSe of the microbiota isolated from the biopsies of the control participants (A) and IBD patients (B).

## Data Availability

Data on the biochemical identification of participants’ medical history are available in Al Noor Specialist Hospital, Ministry of Health, Makkah.

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
