# Peer review of "Impact of Microbiota Diversity on Inflammatory Bowel Disease"

_microorganisms, 2025, doi:10.3390/microorganisms13040710_

Round 1
Reviewer 1 Report
Comments and Suggestions for Authors
This interesting work by Alzahrani et al. evaluated the effects of IBD on microbiota diversity in 60 biopsies using the 16S rRNA method, followed by metagenomic and bioinformatics analyses.
The data are well presented and structured, and the inclusion of numerous images enhances the readability of the study.
Three considerations:
- There are various methods for studying the intestinal microbiota, each with its own limitations and strengths. This aspect should be further explored in the discussion, particularly in relation to the method used in this study and how it compares to others.
- In the introduction, I would include a brief mention of IBD pathogenesis, with a specific focus on the intestinal microbiota and emerging aspects such as the melanocortin system.
- Additionally, it would be useful to include some clinical and demographic data on the patients from whom the biopsies were taken (e.g., years since IBD diagnosis, ongoing and past therapies, current disease activity), as these factors clearly impact the microbiota.
Reviewer 2 Report
Comments and Suggestions for Authors
- Lack of Innovation
The novelty of this study is insufficient, as previous research has already analyzed the gut microbiota in both healthy individuals and patients with inflammatory bowel disease (IBD) with larger sample sizes, providing stronger evidence. Relevant studies include:
- Prideaux, Lani, et al. "Impact of ethnicity, geography, and disease on the microbiota in health and inflammatory bowel disease." Inflammatory Bowel Diseases 19.13 (2013): 2906-2918.
- Sepehri, Shadi, et al. "Microbial diversity of inflamed and noninflamed gut biopsy tissues in inflammatory bowel disease." Inflammatory Bowel Diseases 13.6 (2007): 675-683.
- Uneven Sample Distribution
The sample distribution is unbalanced. Firstly, the number of healthy controls (HCs) is twice that of IBD patients. Additionally, the study does not specify the exact number of samples from Crohn’s disease (CD) and ulcerative colitis (UC) patients. The study states: "A total of 60 intestinal biopsies were collected, comprising 19 patients with CD or UC and 41 HCs." However, the proportions of CD and UC cases remain unclear. - Lack of In-depth Discussion
The discussion section lacks depth and should provide a more detailed comparison of the gut microbiota differences between CD and UC patients. Furthermore, the study should elaborate on how these microbiota variations contribute to the distinct pathogenesis of CD and UC, thereby strengthening the clinical relevance of the findings.

Reviewer 3 Report
Comments and Suggestions for Authors
Comments on microbiome paper
The authors have studied differences in microbiota from IBD (CD or UC) versus healthy people, by taking biopsies during colonoscopy. Biopsies from 19 IBD and 42 healthy people were studied. The manuscript is in general clearly and well written. There are some issues that need to be addressed, as mentioned below.
Introduction is well described and provide the necessary information.
Method section:
- I understand from the manuscript that only one sample was taken from each patient. Would microbiota be affected by the preparation for colonoscopy? Such a limitation should be stated in the manuscript.
- You need to state at which site within the colon was the biopsy taken. It may vary depending on the site. For the IBD patients, would sampling at the inflammatory site differ from a healthy site?
- You need to state how the sampling was done.
- The composition of the ATL buffer needs to be described.
- The concentration of Proteinase K should be stated and its source.
- What was the average incubation time for complete lysis (written until completely lysed).
- I think "100 Am" should be "100 mA". I am not sure you need to explain how the agarose electrophoresis was performed. But you can state the percentage of agarose gel and the composition of the running buffer.
- Did you do 16S gene amplification on purified 16S rRNA or the entire DNA sample. Please state.
- I think it should be "BGI Genomics".
- OTU should be spelled out first time mentioned.
- How much DNA did you extract from each sample?
Result section:
- Please describe the meaning and differences of the Chao, exponent, Shannon, Simpson, ACE, and Good’s coverage indices. This can be done in the Method section with respective formula.
- The definition of evenness and species diversity should be provided so it will be easier to understand the sentence: "greater species evenness indicates greater species diversity". The calculations of these parameters should be stated in the method section.
- In general, the p values are above 0.05, making these changes insignificant.
- The individual data ought to be added to Figure 2.
- Figure legend 2: ACE should be in capital letters.
- Figure 3: Since the letters in A are very small and not readable , I would specify which are IBD and which are HC.
- There is a need to describe how the two principal components were calculated.
- Figure 3B. There are 5 IBD patients that are outstanding. Did they have different disease severities? Are they CD or UC patients?
- Figure 3D: You need to state in the legend that the samples in C are the ones presented in D. It seems that the outstanding 6 samples had lower diversity. Is it so?
- The calculations of UPGMA should be explained in the method section.
- The meaning of the percentage change according to Figure 3b, should be described. In same paragraph: Please describe what you intend by: "possible factors". The factors/parameters in Figure 3b should be stated. And what are the factors that caused the appearance of 6 of the IBD patients.
- Figure 5: Maybe be consistent with the color use for A and B. (The same for all figures).
- Figure 5: FDR should be defined, the p value of * and ** should be stated. These appear for very low abundance species – and cannot been seen in the current graph. In order to convince the reader, I would suggest preparing separate bar graphs for the low abundant species in order to visualize these.
- Figure 5: The individual data should also be included in the figure to show variance. Standard deviation should be included. The same comments for Figure 6.
- Section 3.3.3. Please confirm text here with previous text. In one place it says that the IBD has greater diversity, while here to has lower diversity. Statistics should also be added to the text.
- Figure 7: In legend it is written A and B, but in the figure you gave A-H. It is difficult to know what is what. And the meaning of the bold circles should be described in the legend.
- Section 3.3.4: The differences should be described in the text.
- Figure 8A: The 5 outstanding IBD samples showed increased expression of Propionibacterium and Corynebacterium. Is it possible to make the heatmaps with IBD on one side and healthy control on the other side? In the current presentation they are scattered mixed. Comparison between these 5 and the other IBD patients should be made. A comparison between CD and UC should be made.
- For all the supplementary data, the meaning of the numbers should be stated in the excels.
Discussion:
- A p value of 0.00 is stated. Seems not to be accurate.
Conclusion is more like an anticipation according to what is known in the literature. A specific conclusion to what you can draw from this study should be stated.